# Exploring the effects of added sugar labels on food purchasing behaviour in Australian parents: An online randomised controlled trial

**Devorah Riesenberg**[1¤], **Anna Peeters**[1], **Kathryn Backholer**[1], **Jane Martin**[2], **Cliona Ni Mhurchu**[3,4], **Miranda R. Blake**[1]*

**1** Global Obesity Centre, Institute for Health Transformation, Deakin University, Geelong, Victoria, Australia, **2** Obesity Policy Coalition, Cancer Council Victoria, Melbourne, Victoria, Australia, **3** National Institute for Health Innovation, School of Population Health, The University of Auckland, Auckland, New Zealand, **4** The George Institute for Global Health, Newtown, New South Wales, Australia

¤ Current address: Cancer Council New South Wales, Woolloomooloo, New South Wales, Australia
* miranda.blake@deakin.edu.au

**Data Availability Statement:** The data underlying the results presented in the study are available from Open Science Framework: osf.io/q3b4a.

## Abstract

### Background

Evidence of the effects of front-of-pack added sugar labelling remains limited, especially for foods other than sugary drinks. More information is needed about which labels are likely to be most effective in reducing intended purchases of products with higher added sugar content in realistic contexts to inform policymakers' decisions.

### Objective

To determine the impact of added sugar labels on intended purchases of high sugar breakfast cereals, yoghurt, and non-alcoholic beverages.

### Methods

Australian parents who were regular purchasers of relevant product categories completed an online parallel randomised controlled trial from 31 August 2020 to 13 February 2021. Participants selected their intended purchase from 10 products in each of packaged beverages, breakfast cereal, and yoghurt categories after randomisation to one of seven added sugar labelling conditions in current use or under consideration by the Australian Government. Logistic regressions assessed differences between intervention and control conditions in the odds of intended purchases of a high sugar product.

### Results

2825 eligible participants were randomised with 2582 valid surveys analysed (Control n = 367; 'Nutrition Information Panel (NIP) with Added Sugar' n = 364; 'Teaspoons of Sugar' n = 369; 'Warning' n = 371; 'Health Star Rating (HSR) using Total Sugar' n = 368; 'HSR with Added Sugar' n = 371; 'Sugar in the Ingredients List' n = 372). No consistent effects were found on intended purchases of high sugar products overall or within product categories for

**Funding:** This project was funded by the Deakin University Faculty of Health Health reseArch Capacity building grant scHeme (HAtCH) scheme (MRB). https://www.deakin.edu.au/ The funders had no role in study design, data collection and analysis, decision to publish, or preparation of the manuscript. All authors are researchers within the National Health and Medical Research Council (NHMRC) funded Centre of Research Excellence in Food Retail Environments for Health (RE-FRESH) (APP1152968). NHMRC did not fund the research directly but it provides backbone funding to the research group (for training, communications, etc). The opinions, analysis, and conclusions in this paper are those of the authors and should not be attributed to the NHMRC.

**Competing interests:** I have read the journal's policy and the authors of this manuscript have the following competing interests: Professor Cliona Ni Mhurchu is a member of the trans-Tasman Health Star Rating Advisory Committee (HSRAC) and the Food Standards Australia New Zealand Social Sciences & Economics Advisory Group (SSEAG). Neither HSRAC nor SSEAG had any role in the study design, data collection and analysis, decision to publish, or preparation of the manuscript.

any of the tested labels compared to controls (overall, 'NIP with Added Sugar': OR 1.00 [95%CI 0.83,1.20]; 'Teaspoons of Sugar': 0.94[0.80,1.11]; 'Warning': 1.10[0.93,1.30]; 'HSR with Total Sugar': 1.01[0.85,1.21]; 'HSR with Added Sugar': 1.09[0.92,1.30]; 'Sugar in the Ingredients List': 1.01[0.85,1.21]).

## Conclusions

Findings reinforce the importance of ensuring nutrition labelling policies are introduced as part of a suite of interventions to influence both consumer and manufacturer behaviour.

## Trial registration

Australian New Zealand Clinical Trials Registry, ACTRN12620000858998. Registered 28 August 2020, https://anzctr.org.au/Trial/Registration/TrialReview.aspx?ACTRN=12620000858998.

## Introduction

Free sugars are ubiquitous in the food supply, occurring naturally in many foods as well as being added by food manufacturers [1]. Free sugars, particularly in the form of sugar-sweetened beverages, provide little nutritional value and increase risk of noncommunicable diseases, including obesity, cardiovascular disease, diabetes, and certain cancers [2, 3]. 'Free' sugars include 'added' sugars—sugars added to foods and beverages by the manufacturer, cook, or consumer—plus sugars naturally present in honey, syrups, and fruit juices and fruit juice concentrate [1]. Free sugars are distinct from 'intrinsic' sugars which are present in whole fruits and vegetables, and 'milk' sugars present in dairy products, for which there is minimal evidence of adverse outcomes [4]. Over half of Australians (52%) [5] exceed the World Health Organization (WHO) recommendation for added sugars of <10% of energy intake [1].

The WHO has repeatedly recommended nutrition labelling as part of a suite of measures to prevent non-communicable disease [6]. Over 30 governments globally have introduced a Front-of-Pack (FOP) nutrition label [7] including, (i) display of individual nutrients of concerns such as saturated fat, added sugar, and sodium; (ii) warning labels, where the FOP label is present if the product has high content of a nutrient of concern; and (iii) summary labels, such as a tick or stars [8]. As most shoppers spend a few seconds at most examining food labels before making a decision, labels must be clear and easy to understand [9]. While there is empirical evidence that increased nutrition knowledge increases dietary quality [10], there is no current requirement for packaged foods in Australia to display added sugars content, either in the mandatory nutrition information panel (NIP) on the side or back-of-pack (BOP), or as a voluntary FOP label. Evidence suggests that the NIP in its current form does not influence individual dietary quality [11].

Research has demonstrated that FOP nutrition labels outperform other nutritional labels in improving consumers' ability to find and use nutritional information in purchasing decisions [12–17]. FOP labels can help consumers identify healthier products and improve the healthiness of their hypothetical choices in online surveys [8, 10, 15–18]. A recent systematic review revealed that there is moderate to strong evidence from laboratory trials that FOP added sugar warning labels reduce intended purchasing of less healthy beverages [15]. The review found that graphic warning labels on sugary drinks were more effective than text-based labels at

reducing preference for, and intention to purchase, a sugary drink. Evidence of the effects of FOP added sugar labelling for foods other than sugary drinks and differential effects across food product categories remains limited. There is also limited evidence from online randomised control trials (RCT) that have used realistic label sizing without artificially drawing attention to FOP labels.

Added sugar labelling options are under active consideration by several countries. Food Standards Australia New Zealand (FSANZ) is currently reviewing different FOP and BOP added sugar labelling options [19]. To inform policy makers' decisions, more information is needed about which labels are likely to be most effective in reducing intended purchases of products with higher added sugar content in realistic contexts.

This study aimed to determine the impact of a range of added sugar labels on intended purchase of selected high sugar breakfast cereals, yoghurt, and non-alcoholic packaged beverages. We hypothesised that there would be no difference between intervention label conditions and control conditions in the intended purchases of high sugar products. Our findings supported this null hypothesis, with no consistent differences found in purchases of high sugar products across food product categories for any label tested.

## Materials and methods

### Trial design

The study was a parallel online RCT. Online RCTs are useful for testing policies that are not yet implemented in the real world, and to test the relative effectiveness of different scenarios [20]. The trial protocol was registered with the Australian and New Zealand Clinical Trials Registry (https://anzctr.org.au/Trial/Registration/TrialReview.aspx?ACTRN=12620000858998). See S1 File Application for ethical approval and S2 File Protocol and analysis plan. This study is reported as per the Consolidated Standards Of Reporting Trials (CONSORT) guideline (S3 File CONSORT Checklist). The original study design planned for two trials, using sample size calculations derived from a similar prior study [21], with 'Trial 1' representing the procedure described in this publication, and 'Trial 2' planned to test the effects of displaying multiple labels at once. Recalculated power calculations following initial Trial 1 recruitment indicated a larger sample size would be required for adequate power (see Sample Size), so the research team made the decision to use the remaining resources to focus on additional participant recruitment for what was originally planned as 'Trial 1'.

### Participants

Eligible participants (i) were Australian residents aged ≥18 years; (ii) had access to a computer and internet connection; (iii) were regular (at least once a month) purchasers of non-alcoholic pre-packaged beverages, breakfast cereal, and yoghurt for themselves or their household; (iv) completed at least one supermarket shop per month for their household; and (v) lived with at least one child aged <18 years. Participants were recruited through an online panel provider and were rewarded to the equivalent of ≤AU$6 on survey completion.

We set a quota of 50% regular 'high sugar purchasers' and 50% 'low sugar purchasers' to provide a balance of responses between those whom an added sugar labelling policy is predominantly intended to influence, compared to those whose purchases have less scope for improvement. Participants were asked to select products they purchase at least once a month for themselves or their household from a range of generic options in three product categories: non-alcoholic packaged beverages (hereafter 'beverages'), breakfast cereals, and yoghurts and custards (hereafter 'yoghurts'). If a participant selected one or more 'high' sugar product in each category, they were considered a 'high' sugar purchaser. 'High' sugar products were

classified using the UK Nutrient Profiling Model thresholds (foods >6.25g added sugar/100g; beverages >3.13g/100mL) [22].

## Interventions

Eligible participants were randomised to one of seven labelling conditions in Qualtrics survey platform (Qualtrics, Provo, UT). Each condition included three hypothetical purchasing tasks: one each for beverages, breakfast cereals, and yoghurts. Example choice sets for each product category are shown in S4 File Example choice task for each food category (due to copyright restrictions branded food product labels are replaced with generic examples in the publication). Each task consisted of selecting between 10 products, labelled with one of seven labelling options currently used in Australia or internationally or under consideration by the Australian Government (Fig 1). In each choice task, participants could choose not to purchase any option presented. Warning labels were displayed on products high in added sugars only. All other labels were displayed on every product in the choice task. All labelling conditions (except for the control) also included a BOP NIP with added sugar content. The control condition was the status quo NIP without added sugar content and ingredients list on BOP.

The Health Star Rating (HSR) is Australia's voluntary government-endorsed FOP summary nutrient label, which indicates the healthiness of products from 0.5 (least healthy) to 5 (healthiest) stars based on beneficial and risk nutrient content per 100g [23]. We tested two versions of the HSR algorithm. The first HSR algorithm ('HSR with Total Sugar') was under consideration by the Australian government at the time of the RCT (later adopted) with harsher penalisation for the total sugar content of products [24]. The second version of the HSR algorithm ('HSR with Added Sugar') has been advocated for by public health groups and scores foods based on added sugar content, rather than total sugars [24].

## Selection of food products for testing

We examined the relative effectiveness of the different labels for beverages, breakfast cereals, and yoghurts. These food groups make a significant contribution to Australians' energy intake [5] and there is a wide variation in added sugar content within each product range.

Branded food products were used to increase choice task realism for participants. Food product selection for testing was based on (i) brands available at the two major Australian supermarket chains (Coles and Woolworths) online; (ii) a variety of flavours and product subcategories, e.g., cereal flakes, cereals with added fruit etc.; and (iii) providing a range of added sugar contents. Milk drinks were excluded as they are not usually available in the unrefrigerated beverages section, where the hypothetical choice was being made. A full list of included products and their nutritional content is found in S5 File.

## Calculating added sugar content

For the purposes of this study, 'added sugar' was defined using the WHO definition of 'free sugar', provided in the introduction [1]. Labels within the choice experiment referred to 'added sugars', in line with preferred terminology in the Australian Government consultation on added sugar labelling [19].

The AUSNUT food composition database [25, 26] and a standard protocol [26] were used to determine the added sugar content of branded yoghurts and breakfast cereals. If the ingredients list included sources of added sugars only and no other sugars, then the total sugars were considered to be all added sugars. If the ingredients list included a mixture of added sugars and intrinsic sugars then we calculated added sugar content as a percentage of total sugars

Condition A: 'Control'- current NIP without added sugar information (BOP)

| NUTRITION INFORMAITON | | |
|---|---|---|
| Servings per package: 5 | | |
| Serving size:          250ml | | |
| Av. Quantity | Per Serving | Per 100ml |
| Energy | 196KJ | 78KJ |
| Protein | 0g | 0g |
| Fat total | 0g | 0g |
| -          Saturated | 0g | 0g |
| Carbohydrates | 11.5g | 4.6g |
| -          Total sugars | 11.5g | 4.6g |
| Dietary Fibre | 0g | 0g |
| Sodium | 8mg | 3mg |

Condition B: 'NIP with Added Sugar' information (BOP)

| NUTRITION INFORMAITON | | |
|---|---|---|
| Servings per package: 5 | | |
| Serving size:          250ml | | |
| Av. Quantity | Per Serving | Per 100ml |
| Energy | 196KJ | 78KJ |
| Protein | 0g | 0g |
| Fat total | 0g | 0g |
| -          Saturated | 0g | 0g |
| Carbohydrates | 11.5g | 4.6g |
| -          Total sugars | 11.5g | 4.6g |
| -          Added sugars | 11.5g | 4.6g |
| Dietary Fibre | 0g | 0g |
| Sodium | 8mg | 3mg |

Condition C: 'Teaspoons of Sugar' - Pictorial approach to convey the amount of sugars in a serving of food (FOP)

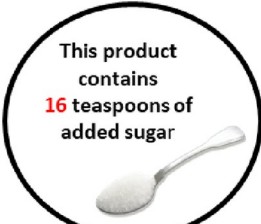

Condition D: 'Warning'- Chilean-style advisory label for foods high in added sugars (FOP)

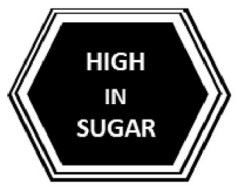

Conditions E and F: 'Health Star Rating with Total Sugar' and 'Health Star Rating with Added Sugar' (FOP)

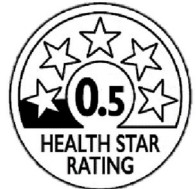

Condition G: 'Sugar in Ingredients List'- Change to statement of ingredients to asterisk ingredients with added sugar (BOP)

Ingredient list: Carbonated water, high fructose corn syrup*, citrate acid, natural flavours, sodium citrate, sodium benzoate

*sugar-based ingredients

**Fig 1. Examples of intervention labels used in the RCT.** BOP, displayed on the Back-of-Pack; FOP, displayed on the Front-of-Pack; NIP, Nutrition Information Panel. Health Star Rating trademarks are owned by the Commonwealth of Australia. Further information on the Health Star Rating can be found at www.healthstarrating.gov.au.

content, based on the percentage added sugar content of a similar generic product in the AUS-NUT database [25].

To calculate the number of teaspoons of sugar per serve, we assumed approximately 4 grams sugar was equivalent to 1 teaspoon. To calculate the modified 'HSR with Added Sugars' using added sugars instead of total sugars, we used a revised points scale based on a dietary target of <10% energy intake from added sugar per day (<52g/day), as proposed during the recent HSR algorithm review [27].

## Survey procedure

Following the RCT, participants completed sociodemographic questions including age, gender, postcode (used to determine area level socioeconomic disadvantage), educational attainment, height and weight (used to determine body mass index (BMI)), and household income and size (used to determined equivalized household income) (S6 File Participant survey questions). Other questions examined participant stated changes in purchasing behaviour under different labelling conditions, understanding of the health risks of sugary drink consumption, and agreement with different communication messages to engage consumers in added sugar labelling policies (using 5-point Likert scales from "strongly agree" to "strongly disagree").

## Outcomes

The primary outcome was the difference between intervention and control conditions in the proportion of participants that intended to purchase a high sugar product, for each product category. The 'no product' intended purchase was grouped with 'low sugar' product intended purchases for analysis. Secondary outcomes included differences in added sugar content (g/100g), and nutrient profile of products using 'HSR with Total Sugar' (from 0.5 to 5), the HSR algorithm later adopted as policy.

## Sample size

A sample size of 2,464 participants (352 participants for each of 7 treatment groups) was estimated to detect a 13% absolute difference in percentage participants selecting a high sugar product between intervention and control conditions ($\beta = 0.80$; $\alpha = 0.0083$; Bonferroni adjustment for comparisons of 6 intervention conditions to control).

## Randomisation and blinding

Assignment of participants to labelling conditions, order in which food product categories were presented, and order of product options within each choice task, were randomised using simple randomisation sequences within the Qualtrics survey platform. It was not possible to blind participants to their label condition. Participants were not aware of what the other label conditions were.

## Statistical analysis

We excluded participants with improbable responses (e.g., reported BMI <15 or >50kg/m$^2$; shopping for >10 children and adults).

We compared participant demographic characteristics across labelling conditions, using chi squared tests. Equivalised household income was calculated by adjusting reported household income (using the mid-point of the reported income brackets) by the number of adults and children in the household [28].

For our primary analysis, we compared the difference in the odds of participants selecting a high sugar product overall and for each product group, between each label condition compared to the control condition, using logistic regression. Secondary continuous outcomes were examined using linear regression for HSR and added sugar content of intended purchases (g/100g). Analyses were adjusted for clustering to take account of repeated participants between categories.

Descriptive statistics compared stated responses to added sugar labelling, understanding of the health risks of sugary drinks, and policy preferences.

Statistical analysis was performed using Stata SE 17.0 (StataCorp, TX USA).

## Ethical approval

Ethical approval was granted by Deakin University Faculty of Health, Human Ethics Advisory Committee (reference number HEAG-H 115_2020). Participants provided informed consent, indicated by clicking the statement "I have read the plain language statement and I agree to participate" before continuing to the online survey.

## Results

2825 participants began the survey from 31$^{st}$ August 2020 to 13$^{th}$ February 2021 (Fig 2) when target sample size was reached. 2582 participants were included in the final analysis (364 to 371 participants per labelling condition). Compared to those included in the analysis, the 243 excluded participants were more likely to be men, have a higher household income, be employed full time, and trying to lose weight (all p<0.05; Table A in S7 File).

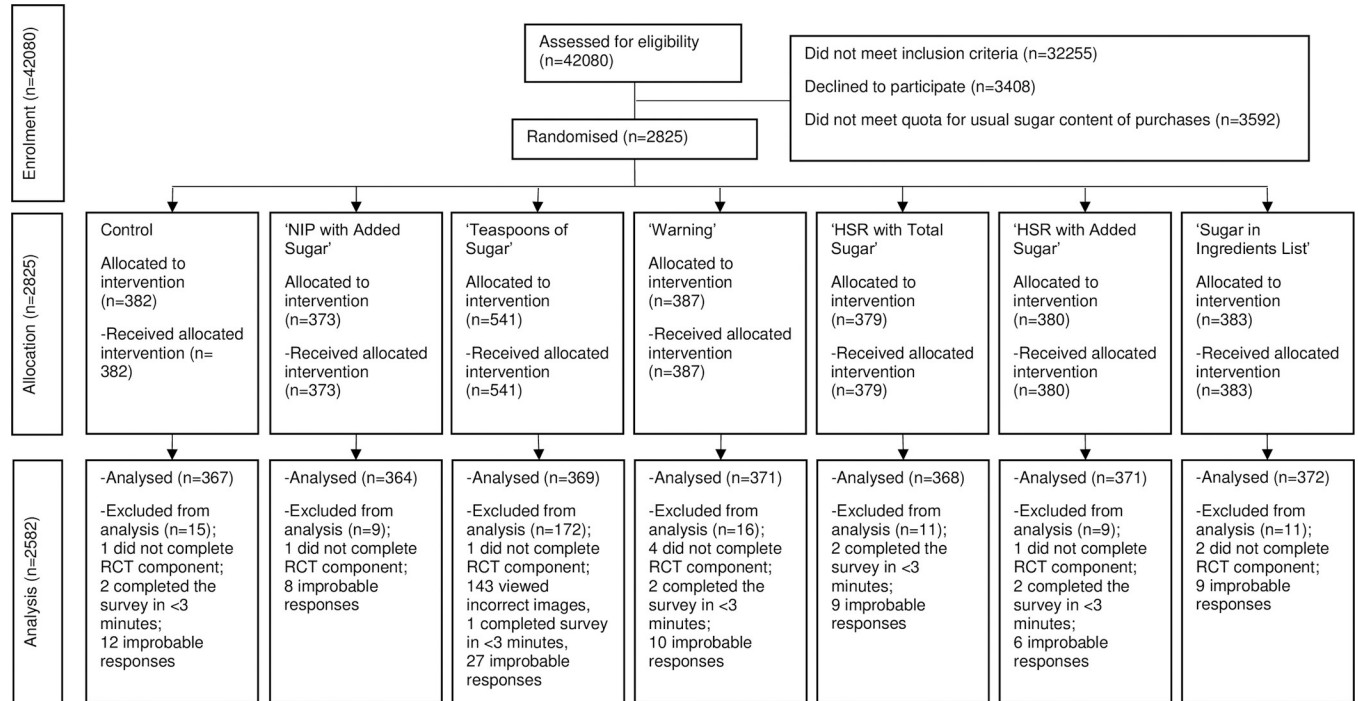

**Fig 2. CONSORT flow diagram.** HSR, Health Star Rating; NIP, Nutrition Information Panel; RCT, Randomised Controlled Trial. 143 subjects who were allocated to the 'Teaspoons of Sugar' labelling condition were assigned to a version with an error in the images and were excluded from the analysis. Additional participants were recruited for this label.

57.8% of included participants were female, 64.0% were aged ≥36 years, 54.3% had attended university, 66.3% lived in an area of low socioeconomic disadvantage, 60.1% were trying to lose weight, and 54.4% reported height and weight equivalent to a BMI ≥25kg/m$^2$ (in the overweight or obese category). No significant differences in demographic characteristics were found between intervention and control conditions (Table 1). Demographic characteristics by sugar content of usual purchases are shown in Table B in S7 File.

Across all conditions, a small proportion of participants elected not to purchase any of the available items in each product category (beverages: 5.32% selections; breakfast cereals: 2.52%; yoghurts: 4.73%). These were grouped with 'low sugar' products for analysis. Five participants selected 'no product' for every product category.

## Effect of added sugar labels on probability of purchasing a high sugar product

Across all product categories combined, 42.0% [95%CI 39.0, 45.1%] of participants in the control condition selected a high sugar product, with the highest proportion of participants selecting a high sugar product for beverages (58.3%; 95%CI 53.3, 63.4%), followed by yoghurts (34.9%; 95%CI 30.0, 39.8%), and breakfast cereals (33.0%; 95%CI 28.1, 37.8%). No significant effects were found for any intervention condition compared to control condition overall (Table 2, full analysis output in Table C in S7 File). In the beverage category, the odds of

**Table 1. Participant demographic characteristics across labelling conditions.**

| Characteristics | n (%) | | | | | | | |
|---|---|---|---|---|---|---|---|---|
| | Total (n = 2582) | Control (n = 367) | NIP with Added Sugar (n = 364) | Teaspoons of Sugar (n = 369) | Warning (n = 371) | HSR with Total Sugar (n = 368) | HSR with Added Sugar (n = 371) | Sugar in Ingredients List (n = 372) |
| **Gender (base: man)** | | | | | | | | |
| Woman | 1471 (57.8) | 205 (56.5) | 201 (56.2) | 235 (64.4) | 209 (57.7) | 206 (56.6) | 217 (59.5) | 198 (53.8) |
| **Age (years) (base: 18 to 35 years)** | | | | | | | | |
| 36 years or older | 1641 (64.0) | 236 (64.7) | 232 (64.3) | 234 (63.6) | 233 (63.8) | 236 (64.5) | 244 (66.1) | 226 (60.8) |
| **Educational attainment (base: less than university)** | | | | | | | | |
| University | 1394 (54.3) | 202 (55.3) | 195 (54.0) | 195 (53.0) | 185 (50.7) | 216 (59.0) | 207 (56.1) | 194 (52.2) |
| **Equivalised household income per week (AUD) (base: <$899) [a]** | | | | | | | | |
| ≥$899 | 999 (38.7) | 144 (39.2) | 141 (38.7) | 130 (35.2) | 144 (38.8) | 144 (39.1) | 154 (41.5) | 142 (38.2) |
| **Employment (base: employed less than full-time)** | | | | | | | | |
| Employed full-time | 1369 (53.4) | 203 (55.6) | 181 (50.1) | 180 (48.9) | 212 (57.8) | 202 (55.2) | 187 (50.7) | 204 (54.8) |
| **Usual sugar purchasing patterns (base: low sugar)** | | | | | | | | |
| High sugar | 1322 (51.2) | 189 (51.5) | 185 (50.8) | 178 (48.2) | 196 (52.8) | 192 (52.2) | 188 (50.7) | 194 (52.2) |
| **Socioeconomic Index for Areas (SEIFA) (base: high socioeconomic disadvantage)** | | | | | | | | |
| Low disadvantage | 1693 (66.3) | 242 (66.3) | 230 (63.9) | 236 (64.1) | 240 (66.1) | 247 (68.2) | 246 (67.4) | 252 (68.1) |
| **Weight loss goals (base: trying to maintain weight or do not have a goal)** | | | | | | | | |
| Trying to lose weight | 1471 (60.1) | 208 (60.6) | 214 (63.1) | 205 (58.9) | 200 (56.5) | 212 (60.2) | 222 (63.1) | 210 (58.2) |
| **Body Mass Index (base: Body Mass Index <25 kg/m$^2$) [b]** | | | | | | | | |
| BMI ≥25 kg/m$^2$ | 953 (54.4) | 135 (54.0) | 129 (52.0) | 124 (51.0) | 137 (54.4) | 136 (54.8) | 152 (58.5) | 140 (55.6) |

HSR, Health Star Rating; NIP, Nutrition Information Panel. There were no significant differences in characteristics between labelling conditions (all p>0.05).

[a] Adjusted for number of adults and children in a household.

[b] 829 participants did not know or elected not to disclose their height and/or weight.

**Table 2. Effect of added sugar labels on selection of a high sugar product, by food category (n = 2582).**

| Label | Odds Ratio [95% CI] | | | |
|---|---|---|---|---|
| | Overall [a] | Product category | | |
| | | Beverages (n = 2582) | Breakfast cereal (n = 2582) | Yoghurts (n = 2582) |
| Base: A: Control | Ref | Ref | Ref | Ref |
| B: Nutrition Information Panel with Added Sugar | 1.00 [0.83, 1.20] | 0.61 [0.46, 0.82] [b] | 1.17 [0.86, 1.59] | 1.40 [1.04, 1.89] [d] |
| C: Teaspoons of Sugar | 0.94 [0.80, 1.11] | 0.73 [0.54, 0.97] [c] | 0.99 [0.73, 1.35] | 1.17 [0.87, 1.58] |
| D: Warning | 1.10 [0.93, 1.30] | 0.88 [0.66, 1.18] | 1.15 [0.85, 1.56] | 1.30 [0.96, 1.75] |
| E: HSR with Total Sugar | 1.01 [0.85, 1.21] | 0.76 [0.57, 1.02] | 1.22 [0.90, 1.65] | 1.13 [0.84, 1.53] |
| F: HSR with Added Sugar | 1.09 [0.92, 1.30] | 0.84 [0.63, 1.12] | 1.35 [1.00, 1.82] | 1.17 [0.87, 1.58] |
| G: Sugar in the Ingredients List | 1.02 [0.85, 1.21] | 0.81 [0.61, 1.09] | 1.19 [0.88, 1.06] | 1.13 [0.83, 1.52] |

HSR, Health Star Rating.

[a] Analysis adjusted for clustering to take account of repeated participants between categories.

[b] p = 0.001

[c] p = 0.032

[d] p = 0.027

selecting a high sugar product were significantly lower than the control condition in the 'NIP with Added Sugar' label (OR: 0.61; 95%CI 0.46, 0.82) and 'Teaspoons of Sugar' label (OR: 0.73; 95%CI 0.54, 0.97) conditions (Table D in S7 File). No significant differences in the odds of selecting a high sugar beverage were found between the control and any other labelling condition.

In the breakfast cereal category, no significant differences in the odds of selecting a high sugar cereal were found between the control and any labelling condition (Table E in S7 File).

In the yoghurt category, the 'NIP with Added Sugar' was associated with higher odds of selecting a high sugar yoghurt, compared to the control condition (OR: 1.40; 95%CI 1.04, 1.89) (Table F in S7 File). No significant differences in the odds of selecting a high sugar yoghurt were found between the control and any other labelling condition.

## Effect of added sugar labelling options on added sugar content of intended purchases

Mean added sugar content of intended purchases in the control condition was 6.44 [95%CI 6.02, 6.85]g/100g. There were no differences between labelling and control conditions in added sugar content of intended purchases, overall (Table G in S7 File).

In the beverage category, the 'NIP with Added Sugar' label and 'Teaspoons of Sugar' label were associated with 2.35 [95%CI -3.92, -0.78]g/100mL and 1.68 [95%CI -3.24, -0.11]g/100mL lower added sugar content of beverage purchases compared to the control condition, respectively. There were no differences in the added sugar content of beverage purchases between the control condition to any other intervention condition (Table H in S7 File).

In the breakfast cereal category, the 'HSR with Added Sugar' condition was associated with 0.38 [95%CI +0.01, +0.75]g/100g higher added sugar content of purchases compared to the control condition. There were no differences in the added sugar content of cereal purchases between the control and any other labelling condition (Table I in S7 File).

In the yoghurt category, no significant differences were found in added sugar content of purchases between the control and any other labelling condition (Table J in S7 File).

### Effect of added sugar labelling options on Health Star Rating of intended purchases

Mean HSR of purchases in the control condition was 3.58 [95%CI 3.52, 3.65]. There was no difference between labelling and control conditions in HSR of intended purchases overall (Table K in S7 File).

In the beverage category, the 'NIP with Added Sugar' label was associated with 0.29 [95%CI +0.07, +0.50] higher mean HSR of purchases compared to the control condition. There were no differences in the HSR of beverage purchases between the control and any other labelling condition (Table L in S7 File).

In the breakfast cereal category, no significant differences were found in the HSR of purchases between any intervention and the control condition (Table M in S7 File).

In the yoghurt category, the 'Sugar in Ingredients List' label was associated with 0.18 [95% CI -0.32, -0.04] lower mean HSR of purchases. There were no differences in the HSR of yoghurt purchases between the control and any other labelling condition (Table N in S7 File).

### Participant stated added sugar labelling and policy preferences

No difference was found between label treatment and knowledge of the risk of any adverse health outcome associated with drinking sugary drinks. The most common minimum number of Health Stars considered healthy was 4.0 (32.8% participants), followed by 3.5 (16.7%) and 3.0 (16.3%).

Most respondents stated they would change their purchases in response to each proposed sugar labelling option (range 67.6% ('Sugar in Ingredients List') to 87.3% ('Teaspoons of Sugar')). For the 'NIP with Added Sugar' (25.3%) and 'Sugar in Ingredients List' (19.5%), the most frequent stated response was to "find a lower sugar alternative". For the 'Warning' label (25.5%), 'Teaspoons of Sugar' (25.6%) and HSR (25.5%) the most frequent response was to "stop purchasing the item" (Table O in S7 File).

Most respondents agreed or strongly agreed with all proposed policy framings including "Government should require stricter standards to ensure that food corporations clearly identify high sugar levels in products" (82.0%), "Consumers need more information to make informed decisions about healthy food products" (81.5%), "We need to set higher standards for how the food industry labels the food we eat" (84.5%), and "We need more nutrition information on food labels so consumers can make the right choices" (82.5%) (Table P in S7 File).

## Discussion

This RCT with 2582 Australian parents found no impact of any of six added sugar labels on the intended purchases of selected high sugar beverages, breakfast cereals or yoghurts, or on added sugar content or HSR of purchases. Significant but inconsistent effects were found for some labels within some food product categories.

Previous evidence on the influence of added sugar labels on food and beverage purchases varies with label content and format. A recent meta-analysis of real-world and hypothetical RCTs and quasi-experimental studies [18] found semi-interpretive labels to be more effective than non-interpretive labels in influencing packaged food choice. In contrast to our findings, large effect sizes were found for warning signs (6 studies; standardized mean differences -0.24 [95%CI -0.35, -0.13]) and teaspoons of sugar labels (2 studies; -0.32 [95%CI -0.48, -0.17]). No effect on intended purchases was found for labels using the guideline daily amount (3 studies) or HSR (2 studies). However, labels were noted to have a greater effect on consumers' understanding of sugar content than on the healthiness of choices. Recent findings suggest that

graphical tobacco-style pictorial labels may be more effective than text- or icon-based warnings among parents [29] and young adults [21]. No jurisdictions currently require such pictorial warning labels, and the current study chose to focus on policy-relevant labels under consideration by the Australian government.

Our findings of inconsistent effect of added sugar labels on sugar content of purchases across different food categories were unexpected. It is unclear from the previous literature how the impact of labels on food choices may differ by food product category. Two real-world 4-week RCTs used smartphone apps to enable study participants to scan product barcodes to receive interpretive or non-interpretive nutrition labels in Australia [30] and in New Zealand [31] and found no effect on healthiness of food purchases in any food category compared to control NIP condition. A recent online RCT with European adults aged 18 to 34 years found an interaction of food label and product category on probability of selecting low, medium, and high sugar products when comparing a traffic light label to pictures of numbers of teaspoons across smoothies, yoghurts and ready-meals categories [32]. This aligns with our study, where some labels improved the healthiness of choices in the beverage category but worsened the healthiness of yoghurt and breakfast cereal choices. These results may be due to random chance or may reflect complex consumer behaviour, and potential unintended consequences of added sugar labelling. For example, the use of a warning label to highlight both yoghurts and custards 'high in added sugar' means that products readily recognised by consumers as less healthy, such as custards, are labelled similarly to products that consumers perceive to be healthy, such as flavoured yoghurts, potentially causing consumer confusion. These findings highlight the need for more in-depth consumer testing of proposed labels, including 'think aloud' techniques to understand the consumer decision-making process [33].

The heterogeneity in effects on intended or actual customer purchases between studies may also relate to the emphasis given to labels, such as absolute and relative label size in laboratory versus real-world settings, other information provided on the pack, and distractions in the choice environment, such as the overwhelming visual and auditory distractions in a supermarket. Our labels were smaller (and arguably more realistic) than have been tested in some previous studies [21]. Similar to our findings, a recent natural experiment in a real-world Dutch supermarket chain found that industry-designed on-shelf labels indicating the sugar content of non-alcoholic beverages using a colour-coded system (blue [lowest sugar content], green, yellow, amber [highest sugar content]) did not affect sales [34].

Most participants stated they would change their purchasing habits in response to each tested labelling condition, in contrast to the results of the RCT. A 2017 report on the implementation of the HSR system including a survey of 2000 Australian consumers [35] found that 43% consumers reported looking at the NIP on all or most food products while at the supermarket. Thirty percent reported they would use HSR to compare products. Other consumer surveys have reported approximately 70% consumers using each NIP and ingredients list when purchasing products for the first time [36]. However, we are not aware of any direct observational data on how frequently consumers use existing labels, or how that information may affect their purchasing decisions in real-world settings. The Starlight RCT in New Zealand found that labels were viewed for 23% of all purchased products, though frequency of viewing decreased over the 4-week RCT [37].

FOP nutrient labelling alone may not provide sufficient incentives for consumer behavioural change in real-world settings. Real-world RCTs with single-label FOP interventions have also failed to find impacts on purchasing [30, 31]. The observed contrast with consumers' anticipated effects and reported desire to select healthier options, may reflect the highly complex consumer choice environment [38]. There are currently multiple strong incentives for consumers to pick less healthy alternatives, for example the visual stimulation of other on-

pack marketing [39]. There is widespread acknowledgment of the need to use multisector and multicomponent interventions to incentivise and support healthier consumer purchasing. In Chile, promising effects on the healthiness of consumer purchases associated with introduction of warning labels for products high in added sugar [40] are likely enhanced by concomitant interventions, including taxes, educational campaigns and advertising restrictions. Additionally, evidence suggests the HSR system introduction may have been associated with manufacturer reformulation in Australia and New Zealand [41]; facilitating healthier intakes without requiring changes in consumer behaviour.

Added sugar on food labels is essential to support these policies on added sugar content, including healthy school food policies, taxation, and advertising restrictions. Evidence [42] and guidance to governments [43] suggest FOP labelling schemes should be accompanied by consumer education campaigns and targeting of population groups of concern including those with low literacy and numeracy. Complementary strategies are needed to target different contexts and consumer subgroups. For example, educational interventions (such as FOP schemes) may be more effective amongst those with higher education levels [44]. Real-world point-of-purchase trials and natural experiments exploring these comprehensive policy approaches are needed to support government decision-making on effective multicomponent approaches to improving dietary intake.

## Strengths and limitations

This study is highly policy-relevant as we tested options for added sugar labelling currently under consideration in Australia and New Zealand [19], focussed on food and beverage purchases of parents, and included policy-relevant and common food product categories. The experiment was limited by the hypothetical nature of the intended purchases. We did not recruit participants to be representative of Australian population demographics, instead targeting adults with children in the home who were regular purchasers of packaged food product categories of interest for added sugar labelling initiatives. Nevertheless, our sample was more highly educated, more likely to be female, and had lower levels of area level disadvantage compared to the general Australian population [45]. The proportion of Australian consumers regularly purchasing high sugar products is currently unknown. This experiment was not powered for stratification by participant demographic characteristics, such as usual sugar purchasing patterns–this would have required almost 5000 participants, which was beyond the project resources.

## Conclusions

This online RCT found no consistent impact of a range of added sugar labels on the intended purchases of selected high sugar packaged non-alcoholic beverages, breakfast cereals, and yoghurts. Evidence suggests labels may be most effective as a part of a suite of interventions to influence consumer and manufacturer behaviour indirectly. Our participants indicated high support for all proposed policy framings for added sugar labelling, indicating that governments are unlikely to face consumer opposition if they take action to introduce added sugar labelling as part of an integrated suite of complementary food policies.

## Supporting information

**S1 File. Application for ethical approval.**
(DOCX)

**S2 File. Protocol and analysis plan.**
(DOCX)

**S3 File. CONSORT checklist.**
(DOCX)

**S4 File. Example choice task for each food category.**
(DOCX)

**S5 File. Included food product options.**
(DOCX)

**S6 File. Participant survey questions.**
(DOCX)

**S7 File. Detailed results.**
(DOCX)

## Author Contributions

**Conceptualization:** Anna Peeters, Kathryn Backholer, Jane Martin, Cliona Ni Mhurchu, Miranda R. Blake.

**Data curation:** Devorah Riesenberg.

**Formal analysis:** Devorah Riesenberg, Kathryn Backholer, Miranda R. Blake.

**Funding acquisition:** Miranda R. Blake.

**Investigation:** Miranda R. Blake.

**Methodology:** Devorah Riesenberg, Anna Peeters, Kathryn Backholer, Cliona Ni Mhurchu, Miranda R. Blake.

**Project administration:** Devorah Riesenberg, Miranda R. Blake.

**Supervision:** Miranda R. Blake.

**Writing – original draft:** Devorah Riesenberg, Miranda R. Blake.

**Writing – review & editing:** Devorah Riesenberg, Anna Peeters, Kathryn Backholer, Jane Martin, Cliona Ni Mhurchu, Miranda R. Blake.

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
