## [Decision Letter · Decision Letter 0]

17 May 2022

PONE-D-22-03985Exploring the effects of added sugar labels on food purchasing behaviour in Australian parents: an online randomised controlled trialPLOS ONE

Dear Dr. Blake,

Thank you for submitting your manuscript to PLOS ONE. After careful consideration, we feel that it has merit but does not fully meet PLOS ONE’s publication criteria as it currently stands. Therefore, we invite you to submit a revised version of the manuscript that addresses the points raised during the review process.

We look forward to receiving your revised manuscript.

Kind regards,

Shahrad Taheri

Academic Editor

PLOS ONE

Journal Requirements:

Reviewers' comments:

Reviewer's Responses to Questions

**Comments to the Author**

1. Is the manuscript technically sound, and do the data support the conclusions?

Reviewer #1: Yes

Reviewer #2: Yes

2. Has the statistical analysis been performed appropriately and rigorously? 

Reviewer #1: Yes

Reviewer #2: Yes

3. Have the authors made all data underlying the findings in their manuscript fully available?

Reviewer #1: Yes

Reviewer #2: Yes

4. Is the manuscript presented in an intelligible fashion and written in standard English?

Reviewer #1: Yes

Reviewer #2: Yes

5. Review Comments to the Author

Reviewer #1: The statistical analysis of the study was well thought out as was the sample size considerations. A sample size of 2,464 participants (352 participants for each of 7 treatment groups) was estimated to detect a 13% absolute difference in percentage participants selecting a high sugar product between intervention and control conditions (power=0.80; α=0.0083; Bonferroni adjustment for comparisons of 6 intervention conditions to control.

The statistical analysis was fairly routine and well planned . The investigators compared participant demographic characteristics across labelling conditions, using chi squared tests. For the primary analysis, they compared the difference in the odds of participants selecting a high sugar product overall and for each product group, between each label condition compared to the control condition, using logistic regression. Secondary continuous outcomes were examined using linear regression for HSR and added sugar content of intended purchases . Analyses were adjusted for clustering to take account of repeated participants between categories.

The results were varied and the analyses of this online RCT showed that the authors found no consistent impact of a range of added sugar labels on the intended purchases of selected high sugar packaged non-alcoholic beverages, breakfast cereals, and yoghurts. Evidence suggests labels may be most effective as a part of a suite of interventions to influence consumer and manufacturer behavior indirectly. The tables and figures were well formatted and easily interpretable.

Reviewer #2: Dear Authors,

I would like to suggest some changes or additions to your paper:

line 57: I wouldn't say that "sugars in all forms provide little nutritional value and increase risk of non communicable diseases (...)" and so on. I think we can agree that a) we need to make a distinction about intrinsic and "free" sugars (as you actually do shortly afterwards, but maybe you can add something more here) and b) it all depends on the equation between caloric needs/portions/frequencies. If it is so, please rephrase and add something short that explains it. If we do not agree on this, please add references for your statement.

Overall I find the results very interesting and I would like to see more emphasis on the importance of not believing that FOPLs alone can somehow be the solution we are looking for. The fact that the statements made by the respondents in my opinion clash with the results obtained, is not trivial and should not be underestimated. It seems that people want to be guided in their choices, except that in practice they choose based on a thousand different variables, ignoring the indications they have asked for. What are the implications for health policy? Perhaps your thoughts on this can help other researchers modify the kinds of questions to be asked or the objects of their research.

Thank you for your work

6. PLOS authors have the option to publish the peer review history of their article (what does this mean?). If published, this will include your full peer review and any attached files.

Reviewer #1: No

Reviewer #2: **Yes: **Claudia Penzavecchia

---

## [Author Response · Author response to Decision Letter 0]

26 May 2022

We thank the editors and reviewers for their helpful comments in improving the manuscript. Below we respond point by point to the reviewer comments. We have made updates to the reference list where required to respond to reviewer comments. 

Journal Requirements:

-Formatting has been updated in line with journal requirements. Note that font and spacing changes have not be track-changed for readability. 

-In the online portal and in Lines 233-6 within the manuscript, ethical approval statement updated: “Ethical approval was granted by Deakin University Faculty of Health, Human Ethics Advisory Committee (reference number HEAG-H 115_2020). Participants provided informed consent, indicated by clicking the statement “I have read the plain language statement and I agree to participate” before continuing to the online survey.”

-Reference list has been reviewed for completeness. References added since original submission, in response to reviewer comments below, are:

4. Makarem N, Bandera EV, Nicholson JM, Parekh N. Consumption of sugars, sugary foods, and sugary beverages in relation to cancer risk: a systematic review of longitudinal studies. Annu Rev Nutr. 2018;38:17-39.

38. Sleddens EFC, Kroeze W, Kohl LFM, Bolten LM, Velema E, Kaspers P, et al. Correlates of dietary behavior in adults: an umbrella review. Nutr Rev. 2015;73(8):477-99.

39. Houghtaling B, Holston D, Szocs C, Penn J, Qi D, Hedrick V. A rapid review of stocking and marketing practices used to sell sugar‐sweetened beverages in US food stores. Obes Rev. 2021;22(4):e13179.

42. Moore SG, Donnelly JK, Jones S, Cade JE. Effect of educational interventions on understanding and use of nutrition labels: A systematic review. Nutrients. 2018;10(10):1432.

43. World Health Organization. Guiding principles and framework manual for front-of-pack labelling for promoting healthy diet. Geneva: World Health Organization. 2019.

44. Backholer K, Beauchamp A, Turrell G, Ball K, Woods J, Martin J, et al. A framework for evaluating the impact of obesity prevention strategies on socioeconomic inequalities in weight. Am J Public Health. 2014;104(10):e43-50.

Reviewer #1: The statistical analysis of the study was well thought out as was the sample size considerations. A sample size of 2,464 participants (352 participants for each of 7 treatment groups) was estimated to detect a 13% absolute difference in percentage participants selecting a high sugar product between intervention and control conditions (power=0.80; α=0.0083; Bonferroni adjustment for comparisons of 6 intervention conditions to control.

The statistical analysis was fairly routine and well planned. The investigators compared participant demographic characteristics across labelling conditions, using chi squared tests. For the primary analysis, they compared the difference in the odds of participants selecting a high sugar product overall and for each product group, between each label condition compared to the control condition, using logistic regression. Secondary continuous outcomes were examined using linear regression for HSR and added sugar content of intended purchases. Analyses were adjusted for clustering to take account of repeated participants between categories.

The results were varied and the analyses of this online RCT showed that the authors found no consistent impact of a range of added sugar labels on the intended purchases of selected high sugar packaged non-alcoholic beverages, breakfast cereals, and yoghurts. Evidence suggests labels may be most effective as a part of a suite of interventions to influence consumer and manufacturer behavior indirectly. The tables and figures were well formatted and easily interpretable.

-We thank the reviewer for their time in reviewing our manuscript. 

Reviewer #2: Dear Authors,

I would like to suggest some changes or additions to your paper:

line 57: I wouldn't say that "sugars in all forms provide little nutritional value and increase risk of non communicable diseases (...)" and so on. I think we can agree that a) we need to make a distinction about intrinsic and "free" sugars (as you actually do shortly afterwards, but maybe you can add something more here) and b) it all depends on the equation between caloric needs/portions/frequencies. If it is so, please rephrase and add something short that explains it. If we do not agree on this, please add references for your statement.

-We agree with the reviewer that the evidence of harm relates mainly to free sugars, and have amended the text to reflect this:

-Amended lines 54-62: “Free sugars, particularly in the form of sugar-sweetened beverages, provide little nutritional value and increase risk of noncommunicable diseases, including obesity, cardiovascular disease, diabetes, and certain cancers (2, 3). ‘Free’ sugars include ‘added’ sugars - sugars added to foods and beverages by the manufacturer, cook, or consumer - plus sugars naturally present in honey, syrups, and fruit juices and fruit juice concentrate (1). Free sugars are distinct from ‘intrinsic’ sugars which are present in whole fruits and vegetables, and ‘milk’ sugars present in dairy products, for which there is minimal evidence of adverse outcomes (4).”

Overall I find the results very interesting and I would like to see more emphasis on the importance of not believing that FOPLs alone can somehow be the solution we are looking for. The fact that the statements made by the respondents in my opinion clash with the results obtained, is not trivial and should not be underestimated. It seems that people want to be guided in their choices, except that in practice they choose based on a thousand different variables, ignoring the indications they have asked for. What are the implications for health policy? Perhaps your thoughts on this can help other researchers modify the kinds of questions to be asked or the objects of their research.

Thank you for your work

-Amended lines 400-423: “FOP nutrient labelling alone may not provide sufficient incentives for consumer behavioural change in real-world settings. Real-world RCTs with single-label FOP interventions have also failed to find impacts on purchasing (30, 31). The observed contrast with consumers’ anticipated effects and reported desire to select healthier options, may reflect the highly complex consumer choice environment (38). There are currently multiple strong incentives for consumers to pick less healthy alternatives, for example the visual stimulation of other on-pack marketing (39). There is widespread acknowledgement of the need to use multisector and multicomponent interventions to incentivise and support healthier consumer purchasing. In Chile, promising effects on the healthiness of consumer purchases associated with introduction of warning labels for products high in added sugar (40) are likely enhanced by concomitant interventions, including taxes, educational campaigns and advertising restrictions. Additionally, evidence suggests the HSR system introduction may have been associated with manufacturer reformulation in Australia and New Zealand (41); facilitating healthier intakes without requiring changes in consumer behaviour. 

-Added sugar on food labels is essential to support these policies on added sugar content, including healthy school food policies, taxation, and advertising restrictions. Evidence (42) and guidance to governments (43) suggests FOP labelling schemes should be accompanied by consumer education campaigns and targeting of population groups of concern including those with low literacy and numeracy. Complementary strategies are needed to target different contexts and consumer subgroups. For example, educational interventions (such as FOP schemes) may be more effective amongst those with higher education levels (44). Real-world point-of-purchase trials and natural experiments exploring these comprehensive policy approaches are needed to support government decision-making on effective multicomponent approaches to improving dietary intake.“

---

## [Editor Report · Decision Letter 1]

1 Jul 2022

Exploring the effects of added sugar labels on food purchasing behaviour in Australian parents: an online randomised controlled trial

PONE-D-22-03985R1

Dear Dr. Blake,

We’re pleased to inform you that your manuscript has been judged scientifically suitable for publication and will be formally accepted for publication once it meets all outstanding technical requirements.

Kind regards,

Shahrad Taheri

Academic Editor

PLOS ONE
---

## [Editor Report · Acceptance letter]

11 Aug 2022

PONE-D-22-03985R1 

Exploring the effects of added sugar labels on food purchasing behaviour in Australian parents: an online randomised controlled trial 

Dear Dr. Blake:

I'm pleased to inform you that your manuscript has been deemed suitable for publication in PLOS ONE. Congratulations! Your manuscript is now with our production department. 

Kind regards, 

on behalf of

Dr. Shahrad Taheri 

Academic Editor

PLOS ONE